

# Amendments to saline-sodic soils showed long-term effects on improving growth and yield of rice (*Oryza sativa* L.)

Dandan Zhao[1,2], Zhichun Wang[1], Fan Yang[1], Wendong Zhu[1,2], Fenghua An[1], Hongyuan Ma[1], Tibor Tóth[3], Xu Liao[1], Hongtao Yang[1] and Lu Zhang[1]

[1] Northeast Institute of Geography and Agroecology, Chinese Academy of Sciences (CAS), Changchun, China
[2] University of Chinese Academy of Sciences, Beijing, China
[3] Research Institute for Soil Science and Agricultural Chemistry of the Hungarian Academy of Sciences, Budapest, Hungary

## ABSTRACT

**Background:** Saline-sodic soils are widely distributed in arid and semi-arid regions around the world. High levels of salt and sodium inhibit the growth and development of crops. However, there has been limited reports on both osmotic potential in soil solutions ($OP_{ss}$) and characteristics of $Na^+$ and $K^+$ absorption in rice in saline-sodic soils under various amendments application.

**Methods:** A field experiment was conducted between 2009 and 2017 to analyze the influence of amendments addition to saline-sodic soils on rice growth and yield. Rice was grown in the soil with no amendment (CK), with desulfurization gypsum (DG), with sandy soil (SS), with farmyard manure (FM) and with the mixture of above amendments (M). The osmotic potential in soil solution, selective absorption of $K^+$ over $Na^+$ (SA), selective transport of $K^+$ over $Na^+$ (ST), the distribution of $K^+$ and $Na^+$ and yield components in rice plants were investigated.

**Results:** The results indicated that amendments application have positive effects on rice yield. The M treatment was the best among the tested amendments with the highest rice grain yield. M treatment increased the $OP_{ss}$ values significantly to relieve the inhibition of the water uptake by plants. Additionally, the M treatment significantly enhanced $K^+$ concentration and impeded $Na^+$ accumulation in shoots. SA values were reduced while ST values were increased for all amendments.

In conclusion, a mixture of desulfurization gypsum, sandy soil and farmyard manure was the best treatment for the improvement of rice growth and yield in the Songnen Plain, northeast China.

# INTRODUCTION

Soil salinity-sodicity is one of the main impediments for crop productivity and sustainability in arid and semiarid areas (*Suarez, 2001*; *Qadir, Noble & Schubert, 2006*). Saline-sodic soils comprise approximately $3.67 \times 10^7$ ha, and Songnen Plain is one of the major saline-sodic areas in China (*Yao, 2008*; *Yang et al., 2016*). pH stress and $Na^+$ toxicity

Corresponding authors
Zhichun Wang,
wangzhichun@iga.ac.cn
Fan Yang, yangfan@iga.ac.cn

are the main causes of the degradation in saline-sodic soils (*Gharaibeh, Eltaif & Shra'Ah, 2010*). Efforts have been made to ameliorate saline-sodic soils including desulfurization gypsum, farmyard manure, sand, hydraulic engineering and phytoremediation (*Qadir et al., 2007*; *Wang, Bai & Yang, 2012*; *Ahmad et al., 2013*).

Desulfurization gypsum provides a sources of $Ca^{2+}$ to replace exchangeable $Na^+$, thereby improving the physical condition of the soil and increasing water infiltration (*Oster, 1982*; *Wang, Bai & Yang, 2013*; *Wang & Yang, 2018*). Manure application improves soil structure and alleviates soil sodicity (*Yu et al., 2010*). Sanding to saline-sodic soils changes soil compactness and reduces salt content (*Wang et al., 2010a*). These amendments showed various improvements of saline-sodic soil properties in practice.

Crops respond to salinity and sodicity in two phases: (1) a continuous osmotic phase that occurs when the potential energy of the saline-sodic soil solution is lowered by its osmotic pressure, thus inhibiting the water uptake of plants; and (2) a slower ionic phase due to ion toxicity or ion imbalance as plants accumulate salt ions over a period of time (*Munns & Tester, 2008*). Most amendment studies focused on soil physiochemical properties (*Chi et al., 2012*; *Zhao et al., 2018*) rather than on the osmotic potential in the soil solution and the selective absorption of ions by plants, although they have important effects on crop biomass (*Wang et al., 2009*).

Rice showed moderate sensitivity to salinity and sodicity (*Maas & Hoffman, 1977*). *Kelly & Rengasamy (2006)* showed that osmotic stress is one of the major factors in reducing crop yield. The decreasing the osmotic potential of the soil solution was inhibitory to the water uptake of plant roots (*Duarte & De Souza, 2016*). The survival of rice plants under saline-sodic conditions is correlated with $Na^+$ and $K^+$ accumulations in plant tissues (*Song & Fujiyama, 1996*). *Yamanouchi, Maeda & Nagai (1987)* found that $Na^+$ concentrations in shoots are inversely correlated with the relative plant growth and yield. The susceptibility of rice plants to salinity and sodicity stress is due to the limited ability to restrict $Na^+$ transportation to shoots (*Matsushita & Matoh, 1991*). This $Na^+$ restricts $K^+$ uptake and $K^+$ is an essential macronutrient for the growth of plants and cannot be substituted by $Na^+$ (*Bhandal & Malik, 1988*). The ability of plants to keep a high cytoplasmic $K^+/Na^+$ ratio is one of the most important mechanisms of salt tolerance (*Maathuis & Amtmann, 1999*).

In this study, we measured the osmotic potential in the soil solution, characterized $K^+$ and $Na^+$ absorption of rice, $K^+$ and $Na^+$ concentrations in shoots and roots, selective absorption/transport for $K^+$ over $Na^+$, distribution of $K^+$, $Na^+$ in rice organs and yield of rice under various soil treatments, including chemical treatment (desulfurization gypsum, DG), physical treatment (sandy soil, SS) and organic treatment (farmyard manure, FM) as well as mixed treatment (M) in saline-sodic soil for planting rice in field. We hypothesized that (1) amendments would increase the osmotic potential in soil solutions; (2) amendments would alter the ion selective absorption and selective transport in saline-sodic soils and (3) the grain yield of rice would be highest by M application according to the synergy among treatments when they applied together in the Songnen Plain, northeast China.

## MATERIALS AND METHODS

### Location description

The study was conducted from 2009 to 2017 at Da'an Sodic Land Experiment Station (45°35′58″–45°36′28″N, 123°50′27″–123°51′31″E, 132.1 m.a.s.l. (above sea level)), operated by the Chinese Academy of Sciences. The climate of this region is semi-humid to semi-arid continental monsoon. The annual mean air temperature is 4.7 °C and the mean annual precipitation of this area is approximately 400–500 mm, and 80% or more of the precipitation occurs between May and September.

The soil at this study site is classified as clay loam with montmorillonite as a dominant mineral. The soil prior to the start of the experiment represents a typically severe saline-sodic soil with pH (1:5 $H_2O$) of 10.47, electrical conductivity (EC) (1:5 $H_2O$) of 2.36 mS $cm^{-1}$, soil organic C (SOC) of 2.80 g $kg^{-1}$ and exchangeable sodium percentage at 79.7% in the top 20 cm soil layer, which is considered to be the effective rooting zone. The main soluble cation was $Na^+$, while the anions were $HCO_3^-$ and $CO_3^{2-}$. Based on the World Reference Base for Soil Resources (*IUSS Working Group WRB, 2014*), the main soil type was classified as solonetz.

### Field design and treatments

The experiment was arranged in a random block design with three replicates of 20 $m^2$ for each plot. There were five treatments: (1) CK, without amendment application; (2) DG, amended with desulfurization gypsum (containing 93% $CaSO_4 2H_2O$) at 3 kg $m^{-2}$; (3) SS, amended with sandy soil at 6 kg $m^{-2}$; (4) FM, amended with 6 kg $m^{-2}$ farmyard manure (5) M, amended with the mixture of desulfurization gypsum, sandy soil and farmyard manure, the amounts of which are equal to those in the DG, SS and FM treatments. Some essential properties of the amendments used in the present study are presented in Table 1 (*Luo et al., 2018*). Plastic cloth buried between plots to a depth of 1 m soil separated plots to prevent disturbance of lateral movement of amendments, water and salt.

The soil amendments were only applied once before the start of this experiment in the late autumn, 2009. The soil amendments were mixed with the 0–20 cm soil layer by rotary cultivator and then irrigation was carried out after 24 h. The CK was also treated by the same method except for the amendment. Agronomic and fertilizer management practices for rice cultivation were the same in all plots and were in accordance with the prevalent system of agriculture in this area. Chemical fertilizers were broadcast over the soil annually at rates of 207 kg N $ha^{-1}$ (as urea containing 46% N), 78 kg P $ha^{-1}$ (as calcium super phosphate containing 12% $P_2O_5$) and 60 kg K $ha^{-1}$ (as potassium sulfate containing 45% $K_2O$). The soil was then plowed to mix the fertilizers into the subsoil.

The local rice cultivar (G19) was planted after wet plowing and sinking between May 20 to the end of May every year for the experiment. Rice seed was sown on normal soil in a greenhouse in early April for nursing, and the 40 day seedlings were transplanted into the plots with a fixed planting spacing of 30 × 16.7 cm. Planting space of 30 × 16.7 cm is a common practice to avoid lodging and cultivation of 3–5 seedlings per hill is

**Table 1 Properties of the amendments used in the present study.**

| Property | Desulfurization gypsum | Sandy soil | Farm manure |
|---|---|---|---|
| pH | 7.62 | 8.92 | 8.30 |
| EC (dS/m) | 34.20 | 0.78 | – |
| SOC (g/kg) | – | 4.23 | 263.30 |
| $K^+$ (g/kg) | 1.00 | 0.001 | 13.60 |
| $Na^+$ (g/kg) | 1.59 | 0.008 | 4.11 |
| $Ca^{2+}$ (g/kg) | 265.30 | 0.10 | 7.49 |
| $Mg^{2+}$ (g/kg) | 1.68 | 0.01 | 10.20 |

Note:
EC, electrical conductivity; SOC, soil organic carbon.

recommended in saline-sodic soil in the Songnen plain (*Wang et al., 2010b*). The depth of 3–7 cm standing water was maintained in the paddy through flood irrigation and runoff drainage during the growth stages of rice. The soils were all drained in the middle of September for harvest.

## Measurements

$K^+$ and $Na^+$ concentrations in rice plant were measured by sampling three hills excluding the border hills from each plot on 20 days before harvesting in 2017. The selected rice hills were observed to be representative of the plot. The rice plants were separated into roots, leaves, sheaths and panicles. The roots were thoroughly washed with water to remove the soil particles. Clean roots were used for estimating $Na^+$ and $K^+$ concentrations. Plant samples were dried for 48 h at 80 °C in an air-forced oven. Dried materials were finely grounded using a ball mill. They were then digested using an acid mixture [sulphuric acid: perchloric acid ($H_2SO_4$: $HClO_4$ = 4:1)] (*Mori et al., 2011*). $K^+$ and $Na^+$ concentrations were determined using an atomic absorption spectrometer (GGX-900). $K^+$ and $Na^+$ concentrations in the shoot were calculated from $K^+$ to $Na^+$ concentrations and dry weights of grains, leaves and sheaths, $K^+$ and $Na^+$ concentrations in the whole plant were calculated from $K^+$ to $Na^+$ concentrations and dry weights of grains, leaves, sheaths and roots.

At harvest in October, the following growth and yield data were determined in 2010, 2012, 2015 and 2017: plant height, panicle length, number of grains per panicle, 1,000-grain weight and grain yield (*Zeng & Shannon, 2000*).

To analyze the soil properties as affected by different amendments, soil sampling was performed after harvest of the rice in the November, 2017. All soil samples, obtained from each plot at six depths of 0–10 cm, 10–20 cm, 20–40 cm, 40–60 cm, 60–80 cm and 80–100 cm were dried at 105 °C for 24 h and passed through a 2 mm diameter sieve. Soil samples were analyzed for electric conductivity (EC in dS m$^{-1}$), soluble $K^+$, $Na^+$ and $Ca^{2+}$ using 1:5 soil to water extracts as described by *Sumner (1993)*.

The EC of 1:5 soil to water extracts ($EC_{1:5}$) was determined by DDS-307 conductivity meter (Shanghai Precision Scientific Instrument Co., Ltd., Shanghai, China), the

concentrations in $mmol_c/L$ of $K^+$, $Na^+$ were determined using flame photometry (FP-6410) and the concentration of $Ca^{2+}$ was measured by EDTA titration (*Jackson, 1956*).

The osmotic potential can serve as a good index for evaluating plant response to saline-sodic stress (*De Souza et al., 2012*). In this experiment, we regard the 1:5 soil to water extracts as soil solution, and the osmotic potential in the soil solution (OPss) was calculated as follows:

$$OP_{ss} = (-0.36) \times 10EC \text{ (\textit{Bohn, Myer \& O'Connor, 2002})}$$

## SA and ST calculation

Selective absorption of $K^+$ over $Na^+$ (SA) represents the net capacity of a plant to absorb $K^+$ relative to $Na^+$ from the shallow soil (0–40 cm); Selective transport of $K^+$ over $Na^+$ (ST) reflects the net capacity of a plant to favor transport of $K^+$ over $Na^+$ from the root to shoot (*Wang et al., 2004a*). In this study, SA and ST values were calculated according to the following formula (*Wang et al., 2002*, *2004b*) using data obtained from the experiments described earlier:

SA = (K/Na in root dry weight)/(soil K/Na at 0–40 cm depth)
ST = (K/Na in shoot dry weight)/(K/Na in root dry weight)

## Statistical analysis

Statistical analysis was performed by using the statistical software SPSS 20.0 (New York, USA). We used a randomized block design with three replicates, treated block as a random effect and allowing treatment to enter the model as a fixed effect. One-way analysis of variance (ANOVA) was used for comparing the differences in the means among treatments within each plot. On the basis of the ANOVA results, Duncan's multiple range test (DMRT) was used to determine differences among the amendment treatments. A probability value of $P < 0.05$ was used as the criterion for statistical significance. A comprehensive analysis table shows the results of ANOVAs for the effects of treatment and block on rice plant and soil characteristics (Table S1).

# RESULTS

## Effect of amendments application on osmotic potential in soil solution

The osmotic potential in the soil solution ($OP_{ss}$) was increased by amendments application compared to the control. The amplitude of variation of $OP_{ss}$ was from −4.39 bars in the 80–100 cm soil layer under CK treatment to −1.04 bars in the 10–20 cm soil layer under M treatment. In the 0–40 cm soil layer, amendments application generally increased the $OP_{ss}$ values in the following order: M>DG>SS>FM>CK (Fig. 1). In the 0–10 cm soil layer, the M, DG, SS and FM treatment increased the $OP_{ss}$ by 53.8%, 40.1%, 29.1% and 12.2% compared to the CK treatment, respectively. In the same soil layer, the highest $OP_{ss}$ was observed for M, which means that the ability to reduce the salt concentration of soil solution is strongest, followed by DG.

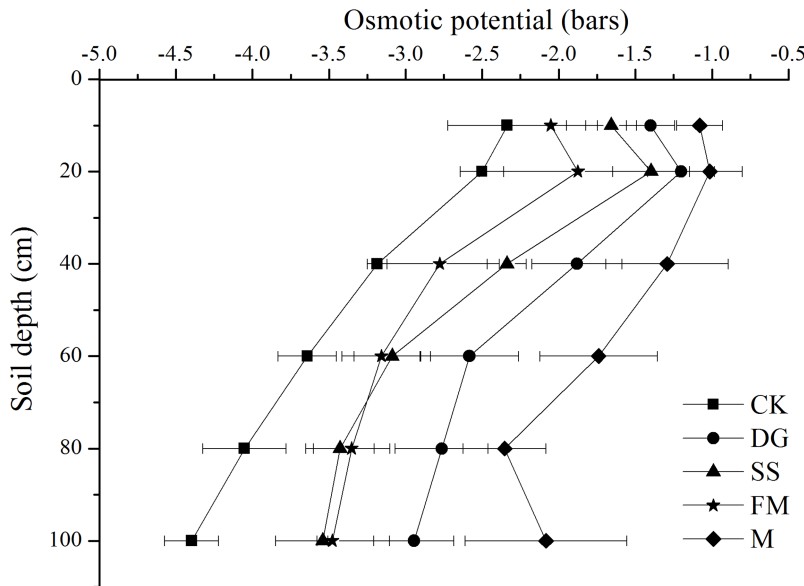

**Figure 1 Osmotic potential of the 1:5 soil water extract at various soil profile depths with different amendments application.** CK, control, without amendments application; DG, desulfurization gypsum; SS, sandy soil; FM, farmyard manure; M, mixture of desulfurization gypsum, sandy soil and farmyard manure. Bars represent the standard error of the mean of three replications.

## Effect of amendments application on Na$^+$ and K$^+$ concentrations in rice shoots and roots

The Na$^+$ concentration in shoots of rice plants varied with different amendments applied in the saline-sodic soil (Fig. 2A). Rice shoots of plants in M treatment showed the lowest Na$^+$ concentration of 0.91 mg/g dry weight and the Na$^+$ concentrations in FM, CK, SS and DG treatments were 4.4%, 7.7%, 8.8% and 11.0% higher than that in M treatment, respectively. The difference in Na$^+$ concentration between DG and M treatments was significant. The mean root Na$^+$ concentration was highest in the CK treatment, and 0.8%, 7.1%, 9.2% and 15.1% lower in M, SS, FM and DG treatments, respectively. However, the differences on Na$^+$ concentration in rice root among amendment treatments and CK were non-significant (Fig. 2B). Amendments application significantly enhanced K$^+$ concentration in rice shoots compared to the control treatment, with the highest K$^+$ concentration found for DG (Fig. 2C). The K$^+$ concentration in rice roots with M, SS, FM treatments were lower than that with the control treatment. The lowest K$^+$ concentration was observed for FM, which was 16.8% lower than that of CK (Fig. 2D).

## Selective absorption and transport of K$^+$ over Na$^+$ in rice plant

Compared to the CK treatment, the M treatment significantly decreased the SA value of the rice by 74.8% (Fig. 3A). However, the M treatment significantly increased the ST value of the rice compared to the CK treatment, which was 1.5 times more than the ST value of the CK treatment (Fig. 3B). Amendment application hindered the uptake of K$^+$ over Na$^+$ from soil to root (SA) compared with CK (Fig. 3A), which is probably a

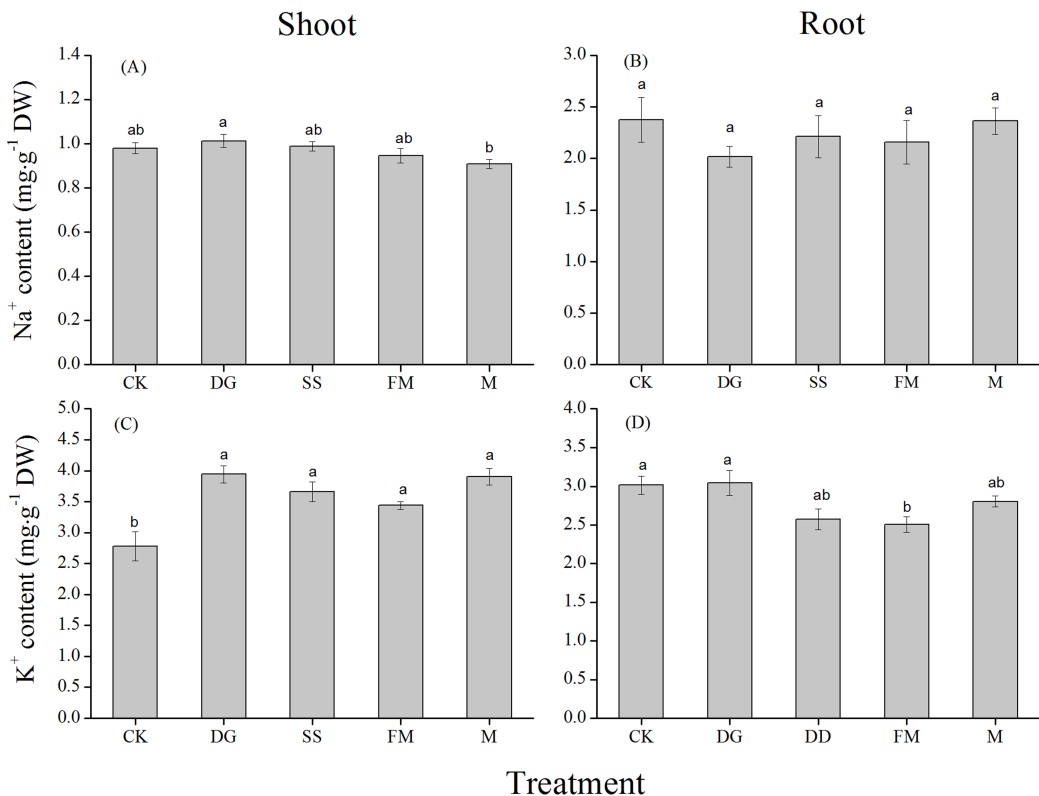

**Figure 2 Na⁺ and K⁺ concentrations in different parts of rice plants with various treatments.**
(A) Na⁺ concentration in rice shoot with various treatments; (B) Na⁺ concentration in rice root with various treatments; (C) K⁺ concentration in rice shoot with various treatments; (D) K⁺ concentration in rice root with various treatments. Shoot, the aboveground part of rice; Root, the underground part of rice. CK, control, without amendments application; DG, desulfurization gypsum; SS, sandy soil; FM, farmyard manure; M, mixture of desulfurization gypsum, sandy soil and farmyard manure. Bars represent the standard error of the mean of three replications. Different letters denote means that are significantly different from each other ($P < 0.05$).

consequence of rice physiological adjustment. Amendment application enhanced the uptake of K⁺ over Na⁺ from root to shoot (ST) compared with CK (Fig. 3B). This was attributed to strong selective transport of K⁺ over Na⁺ under amendment application.

The mean Na⁺ concentration in the soil extracts decreased from a maximum (6.68 mmol$_c$/L) in the CK treatment to 3.16, 4.35, 5.11 and 6.60 mmol$_c$/L with M, DG, SS and FM treatments, respectively. The differences among M and CK were significant (Fig. 4A). Amendment application slightly enhanced the K⁺ concentration in the soil extracts compared to CK (Fig. 4B). The Ca²⁺ concentration in the soil extracts were higher for treatments with amendments than the one without and differences among the four different amendments were not significant (Fig. 4C; Supplemental File 2).

## Characteristics of distribution of Na⁺ and K⁺ in rice with different amendments application

There were little differences in Na⁺ concentration in the whole rice plants among different treatments in the saline-sodic soils in this experiment (Fig. 5A). K⁺ concentration

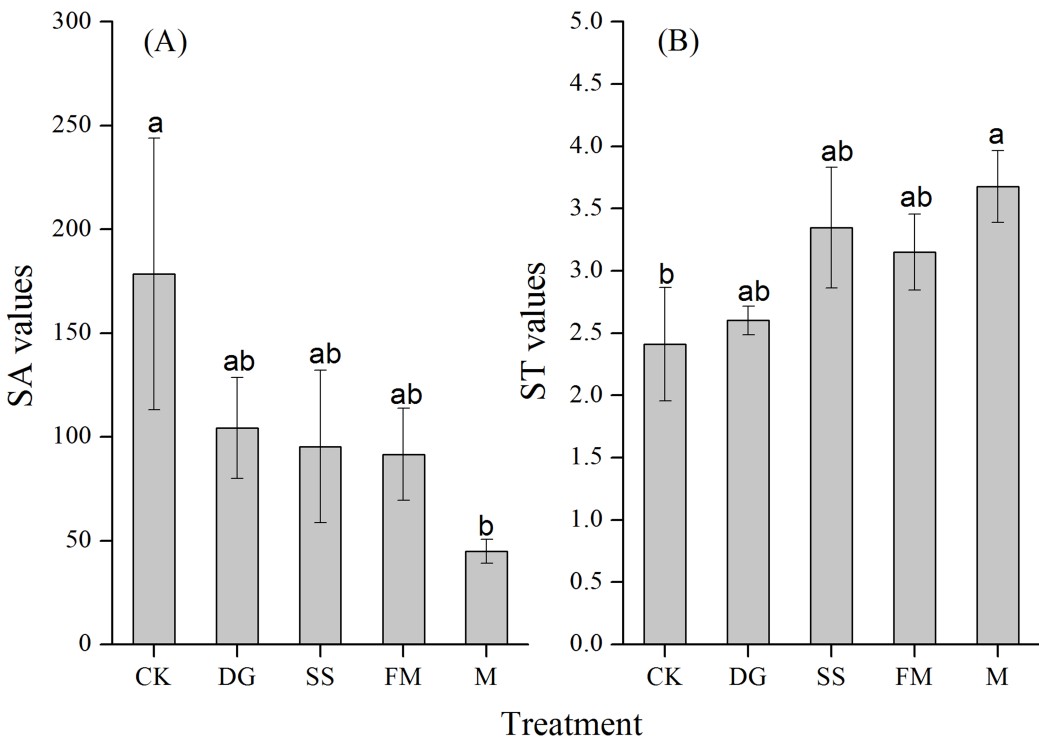

**Figure 3 Selective absorption (SA) and selective transport (ST) of rice with various treatments.**
(A) Selective absorption (SA) of rice with various treatments; (B) Selective transport (ST) of rice with various treatments. SA values, selective absorption of $K^+$ over $Na^+$; ST values, selective transport of $K^+$ over $Na^+$. CK, control, without amendments application; DG, desulfurization gypsum; SS, sandy soil; FM, farmyard manure; M, mixture of desulfurization gypsum, sandy soil and farmyard manure. Bars represent the standard error of the mean of three replications. Different letters denote means that are significantly different from each other ($P < 0.05$).

in the whole plant was significantly enhanced after amendment application, but the differences between the four treatments with amendments were non-significant (Fig. 5B). $Na^+$ absorbed by the whole plant was almost the same with and without amendments, which was different from the observations on rice organs (Fig. 5A; Table 2). The $Na^+$ concentrations in rice roots and grains both decreased when applying amendments in the saline-sodic soils; which were contrary to the rise of $K^+$ concentrations in sheaths and leaves (Table 2). Compared to the control treatment, DG, SS, FM and M treatments increased the $K^+$ concentrations in rice sheaths by 57.2%, 54.9%, 44.1% and 25.5%, respectively.

For the distribution of ions in rice organs, there was a higher proportion of the total $K^+$ in leaves. More $Na^+$ was found in roots. The order of accumulation of $Na^+$ in various organs was roots > leaves > sheaths > grains (Table 2; Supplemental File 1). The order is imposed by the fact that the root system retains more $Na^+$ and prevents $Na^+$ from being transported to the aboveground organs in saline-sodic soils, resulting in higher $K^+$ proportion in leaves, sheaths and grains. This was also illustrated as being beneficial to normal metabolic activity (*Borsani, 2001*; *Ahmad & Jabeen, 2005*).

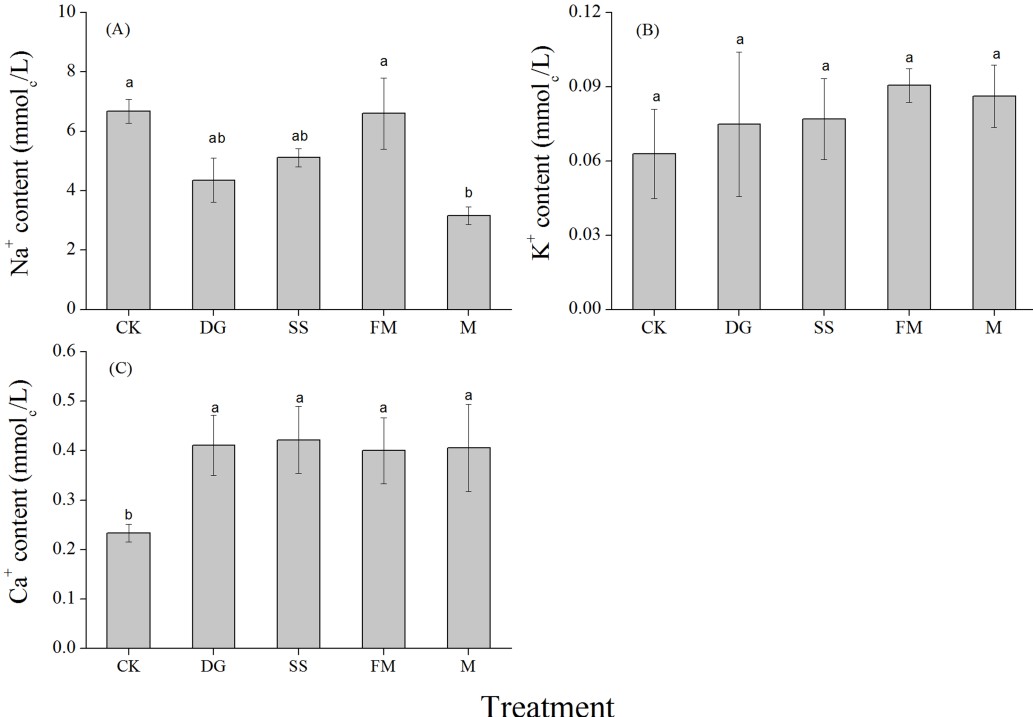

**Figure 4 Na⁺, K⁺ and Ca²⁺ concentrations in the 1:5 soil water extract (0–40 cm) with various treatments.** (A) Na⁺ concentration in the 1:5 soil water extract (0–40 cm) with various treatments; (B) K⁺ concentration in the 1:5 soil water extract (0–40 cm) with various treatments; (C) Ca²⁺ concentration in the 1:5 soil water extract (0–40 cm) with various treatments. CK, control, without amendments application; DG, desulfurization gypsum; SS, sandy soil; FM, farmyard manure; M, mixture of desulfurization gypsum, sandy soil and farmyard manure. Bars represent the standard error of the mean of three replications. Different letters denote means that are significantly different from each other ($P < 0.05$).

## Relationship between OP$_{ss}$, selective absorption and yield of rice

The grain yield of the 4 years of 2010, 2012, 2015 and 2017 were taken as representative of the trend of rice yield from 2009 to 2017 (Fig. 6; Supplemental File 4). In terms of grain yield, the M was the best treatment, next is the DG treatment. The grain yield of rice with amendments application were significantly higher than without amendments application except in 2015.

Amendment treatments significantly enhanced the grain yield of rice compared to the control in 2017 (Table 3). The differences, however, among different amendments were not significant at $P < 0.05$. Soil amendment application generally increased the 1,000-grain weight in the following order: FM > M > DG > SS > CK (Table 3; Supplemental File 3). Additionally, the FM and M treatments significantly increased the 1,000-grain weight to 1.16 and 1.13 times more than the CK treatment, respectively (Table 3). Compared to the CK treatment, the SS treatment considerably enhanced the number of grains per panicle (Table 3). There was no significant difference on rice height and panicle length between various treatments (Table 3).

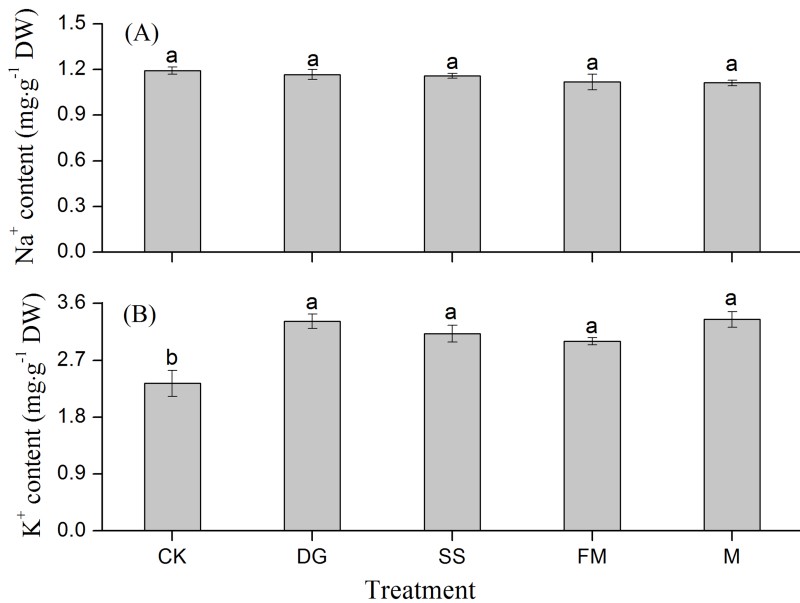

**Figure 5 Na$^+$ and K$^+$ concentrations in the whole rice plant with various treatments.** (A) Na$^+$ concentration in the whole rice plant with various treatments; (B) K$^+$ concentration in the whole rice plant with various treatments. DW, dry weight. CK, control, without amendments application; DG, desulfurization gypsum; SS, sandy soil; FM, farmyard manure; M, mixture of desulfurization gypsum, sandy soil and farmyard manure. Bars represent the standard error of the mean of three replications. Different letters denote means that are significantly different from each other ($P < 0.05$).

Significant positive correlations were found between OP$_{ss}$ in the 0–20 cm soil layer and the 1,000-grain weight ($R^2 = 0.992$, $P < 0.001$, Table 4). Significant negative correlations were found between SA and rice grain yield ($R^2 = 0.925$, $P < 0.05$, Table 4) and between SA and 1,000-grain weight of rice ($R^2 = 0.884$, $P = 0.047$, Table 4). There was no significant correlation between either SA or ST and other growth parameters and yield of rice (Table 4).

# DISCUSSION

## Characteristics of Na$^+$ and K$^+$ absorption in rice

Compared with the CK treatment, the selective absorption of K$^+$ over Na$^+$ (SA) decreased significantly with the M application in this study. When the M applied, osmotic stress and Na$^+$ toxicity were significantly decreased leading to better plant growth in saline-sodic soils (*Swarup, 1982*; *Yuncai & Schmidhalter, 2005*; *Luo et al., 2018*; *Shi et al., 2019*). Similar to our results, previous studies have shown that plants accumulate excessive Na$^+$ in their shoots under stress caused by high salinity-sodicity (*Roy & Mishra, 2014*), and Na$^+$ concentration in shoots increased significantly with a surge in soil salinity-sodicity (*Syed & Abdur, 2017*).

Adding amendments reduces the salinity-sodicity stress of plants growing in the amended soil (*Chaganti & Crohn, 2015*). Therefore, the rice planted in the CK plot was under a higher external salinity-sodicity stress. As a result, the SA value of rice plants

**Table 2 $Na^+$ and $K^+$ concentrations and $K^+/Na^+$ ratios in different organs of rice plant with various treatments.**

| Treatment | Organ | $Na^+$ (mg/g DW) | $K^+$ (mg/g DW) | $K^+/Na^+$ |
|---|---|---|---|---|
| CK | Grain | 0.53c | 2.75ab | 7.07a |
| | Leaf | 1.38b | 3.61a | 2.57b |
| | Sheath | 1.06b | 1.99b | 1.90b |
| | Root | 2.38a | 3.01a | 1.36b |
| DG | Grain | 0.25d | 2.62b | 10.61a |
| | Leaf | 1.63b | 5.06a | 3.18b |
| | Sheath | 1.30c | 4.42a | 3.40b |
| | Root | 2.02a | 3.04b | 1.53c |
| SS | Grain | 0.27d | 2.69c | 10.11a |
| | Leaf | 1.59b | 4.93a | 3.11b |
| | Sheath | 1.19c | 3.56b | 2.93b |
| | Root | 2.21a | 2.57c | 1.28c |
| FM | Grain | 0.19d | 2.60b | 15.15b |
| | Leaf | 1.61b | 5.19a | 3.23b |
| | Sheath | 1.13c | 2.67b | 2.40a |
| | Root | 2.16a | 2.51b | 1.25b |
| M | Grain | 0.24c | 2.61b | 12.59a |
| | Leaf | 1.27b | 4.62a | 3.85b |
| | Sheath | 1.31b | 4.65a | 3.54b |
| | Root | 2.36a | 2.80b | 1.21c |

Notes:
Lowercase letters after data in a column for each treatment indicate that ion contents were significantly different at $P = 0.05$.
CK, control, without amendments application; DG, desulfurization gypsum; SS, sandy soil; FM, farmyard manure; M, mixture of desulfurization gypsum, sandy soil and farmyard manure.

with CK was higher than those with amendments application and maintained a high cytosolic $K^+/Na^+$ ratio. This is thought to be one of the most important mechanisms of salt tolerance exhibited by plants (*Gorham, 1990*; *Dubcovsky et al., 1996*; *Munns & James, 2008*; *Munns et al., 2010*).

## Effects of $Ca^{2+}$ on SA and ST values

The competition between $K^+$ and $Na^+$ to entry into plants can result in significant adverse effects on plants' growth, where concentrations of $Na^+$ often exceed those of $K^+$ (*Tester & Davenport, 2003*). Therefore, the maintenance of a high $K^+/Na^+$ ratio in plants is essential (*Maathuis & Amtmann, 1999*). Amendments of $Ca^{2+}$ promoted $K^+$ over $Na^+$ absorption, resulting in the enhancement of selective absorption of $K^+$ over $Na^+$ (*Alama et al., 2002*). $Ca^{2+}$ can replace $Na^+$ in plants, which restores cell wall stability and plasma membrane integrity (*Zhang, Flowers & Wang, 2010*; *Wu & Wang, 2012*). Although alleviation of $Na^+$ toxicity by supplemental $Ca^{2+}$ was confirmed, the responses varied with different plant species. Under similar saline-sodic conditions, amendments of $Ca^{2+}$ were found to obviously increase $K^+/Na^+$ selectivity of both roots and shoots (SA and ST values) in *Medicago sativa* (*Al-Khateeb, 2006*) and *Cornus sericea* (*Renault & Affifi, 2009*).

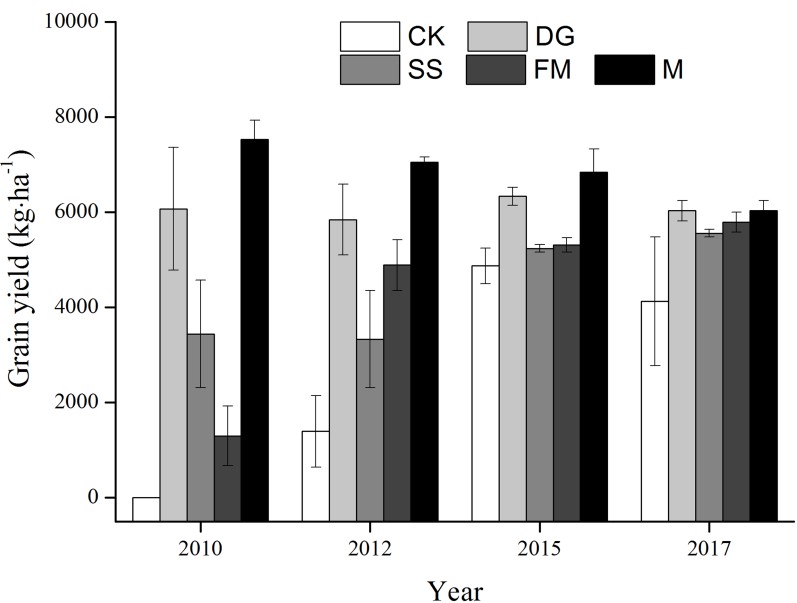

**Figure 6 The trend of grain yield.** CK, control, without amendments application; DG, desulfurization gypsum; SS, sandy soil; FM, farmyard manure; M, mixture of desulfurization gypsum, sandy soil and farmyard manure. Bars represent the standard error of the mean of three replications.

**Table 3 Effects of amendments application on growth and yield of rice plant in 2017.**

| Treatment | Height (cm) | Panicle length (cm) | Number of grains per panicle | 1,000-grain weight (g) | Grain yield (kg/ha) |
|---|---|---|---|---|---|
| CK | 89.0 ± 3.2a | 14.3 ± 0.4a | 59.3 ± 3.9b | 19.4 ± 0.6b | 4,130 ± 1349.2b |
| DG | 87.8 ± 3.4a | 15.3 ± 0.4a | 78.5 ± 9.5ab | 21.7 ± 0.6ab | 6,030 ± 209.9a |
| SS | 89.8 ± 2.1a | 15.3 ± 0.6a | 86.7 ± 5.2a | 20.7 ± 0.8ab | 5,560 ± 79.4a |
| FM | 94.2 ± 1.1a | 14.3 ± 0.3a | 68.9 ± 8.4ab | 22.4 ± 0.7a | 5,790 ± 209.9a |
| M | 92.5 ± 2.6a | 14.3 ± 0.3a | 78.2 ± 4.6ab | 21.9 ± 0.9a | 6,030 ± 209.9a |

**Notes:**
Mean value and its standard error (SE) are reported. Different letters denote means that are significantly different from each other ($P < 0.05$).
CK, control, without amendments application; DG, desulfurization gypsum; SS, sandy soil; FM, farmyard manure; M, mixture of desulfurization gypsum, sandy soil and farmyard manure.

**Table 4 Correlation coefficients among OP, SA, ST values and different growth and yield of rice in 2017.**

| | $OP_{SS}$ (bars) | SA (selective absorption) | ST value (selective transport) | Height (cm) | Panicle length (cm) | Number of grains per panicle | 1,000-grain weight (g) |
|---|---|---|---|---|---|---|---|
| SA (selective absorption) | −0.857 | | | | | | |
| ST (selective transport) | 0.628 | −0.879* | | | | | |
| Height (cm) | 0.695 | −0.589 | 0.278 | | | | |
| Panicle length (cm) | −0.146 | −0.205 | 0.391 | −0.62 | | | |
| Number of grains per panicle | 0.276 | −0.492 | 0.319 | −0.08 | 0.727 | | |
| 1,000-grain weight (g) | 0.992** | −0.884* | 0.671 | 0.619 | −0.024 | 0.375 | |
| Grain yield (kg/ha) | 0.789 | −0.925* | 0.821 | 0.303 | 0.477 | 0.714 | 0.855 |

**Notes:**
* Denote correlation at the 0.05 levels of significance.
** Denote correlation at the 0.01 levels of significance.

This is in contrast with *Wang, Zhang & Flower (2007)*, who reported that amendments of $Ca^{2+}$ had no influence on SA and ST values of *Suaeda maritima*. In addition, the responses of $Na^+$ to $Ca^{2+}$ also varied with osmotic potential in soil solution in the same plant species. In rice, $Ca^{2+}$ did not have significant effects on selective absorption and selective transport of $K^+$ over $Na^+$ of plants when subjected to low osmotic potential in soil solution (*Yeo & Flowers, 2010*). This is consistent with the results obtained in the present study: there were not significant differences among CK, DG, SS and FM treatment on SA and ST values (Fig. 3). In contrast, M application significantly decreased roots $Na^+$ absorption and increased shoots $K^+$ accumulation in rice. It is proposed that the presence of $Ca^{2+}$ could enhance $K^+/Na^+$ selectivity and regulate ion homeostasis in rice under low saline-sidicity condition.

In rice, a minority of the ions reaching the plant shoots are the consequence of leakage along the transpirational bypass flow to the xylem and $Ca^{2+}$ application can reduce the bypass flow of rice (*Faiyue et al., 2010*; *Anil et al., 2005*). This reduction in the bypass flow is positively related with the concomitant reduction in the shoot $Na^+$ uptake (*Anil et al., 2005*). In addition, a majority of the ions reaching the shoots of rice should be transported via the symplast pathway. Therefore, $Ca^{2+}$ plays important role in regulating apoplast and symplast pathways involved in $Na^+$ transport.

### Yield of rice

Transient salinity affects the plants' absorption of available water, which results in a reduction in plant yield (*Rengasamy, 2010a*, *2010b*). However, application of amendments to saline-sodic soils can alleviate the salinity-sodicity stress on plants (*Irshad et al., 2002*). Amendments application in our study enhanced the $OP_{ss}$ values, and then decreased osmotic pressure of the soil solution. This ultimately increased the plant growth and yield of rice in the saline-sodic soils. Applying a small amount of calcium thus was shown to enhance the plants' salt tolerance (*Cramer, 1992*).

DG, SS and FM application are known to improve the root environment and increase rice yield (*Abrishamkesh, Gorji & Asadi, 2015*). In this study, we found that the mixture of DG, SS and FM application significantly reduced the absorption of $Na^+$ in rice shoots and led to the highest rice grain yields, which may be due to the synergistic effect of these three amendments. However, the contribution of each amendment to the rice yield needs to be quantified in future studies.

### CONCLUSIONS

In this field experiment, the amendments application significantly increased the yield of rice. In particularly, the M treatment was the best among the tested amendment treatments, with the highest rice grain yield in the saline-sodic soils, although the differences between amendment treatments were not significant. Relative to the CK treatment, the FM and M treatments significantly enhanced the 1,000-grain weight and the SS treatment significantly improved the number of grains per panicle. All treatments increased the $OP_{ss}$ significantly, thus relieving the inhibition of water uptake by plants. In addition, a positive effect of amendments application on reducing $Na^+$ accumulation

and increasing the uptake of $K^+$ of rice shoot was observed. Amendments application increased ST values and decreased SA values. Moreover, there existed an ion regionalization distribution in rice plant; there was a higher $K^+$ proportion in leaves and a higher $Na^+$ proportion in roots. Collectively, the mixture of desulfurization gypsum, sandy soil and farmyard manure provided excellent results for increasing the yield of rice in the saline-sodic soils in the Songnen Plain, northeast China.

## ABBREVIATIONS

| | |
|---|---|
| $OP_{ss}$ | osmotic potential in the soil solution, bars |
| SA | selective absorption of $K^+$ over $Na^+$ |
| ST | selective transport of $K^+$ over $Na^+$ |
| EC | electrical conductivity in 1:5 soil to water extracts, dS $m^{-1}$ |
| SOC | soil organic carbon, g $kg^{-1}$ |
| CK | control, without amendments application |
| DG | desulfurization gypsum |
| SS | sandy soil |
| FM | farmyard manure |
| M | mixture of desulfurization, sandy soil and farmyard manure |

## ACKNOWLEDGEMENTS

We thank Da'an Sodic Land Experiment Station of China for providing the experimental plot.

### Funding

This work was supported by the National Key Research & Development Program of China (No. 2016YFC0501200), the National Natural Science Foundation of China (Nos. 41571210 and 41771250), the Science-technology Development Initiative of Jilin Province (No. 20180201012SF), and CAS President's International Fellowship Initiative, PIFI. The funders had no role in study design, data collection and analysis, decision to publish, or preparation of the manuscript.

### Grant Disclosures

The following grant information was disclosed by the authors:
National Key Research & Development Program of China: 2016YFC0501200.
National Natural Science Foundation of China: 41571210 and 41771250.
Science-technology Development Initiative of Jilin Province: 20180201012SF.
CAS President's International Fellowship Initiative, PIFI.

### Competing Interests

The authors declare that they have no competing interests.

## Author Contributions

- Dandan Zhao conceived and designed the experiments, performed the experiments, analyzed the data, prepared figures and/or tables, authored or reviewed drafts of the paper, and approved the final draft.
- Zhichun Wang conceived and designed the experiments, authored or reviewed drafts of the paper, and approved the final draft.
- Fan Yang conceived and designed the experiments, authored or reviewed drafts of the paper, and approved the final draft.
- Wendong Zhu performed the experiments, prepared figures and/or tables, and approved the final draft.
- Fenghua An performed the experiments, authored or reviewed drafts of the paper, and approved the final draft.
- Hongyuan Ma analyzed the data, authored or reviewed drafts of the paper, and approved the final draft.
- Tibor Tóth analyzed the data, authored or reviewed drafts of the paper, and approved the final draft.
- Xu Liao performed the experiments, prepared figures and/or tables, and approved the final draft.
- Hongtao Yang performed the experiments, prepared figures and/or tables, and approved the final draft.
- Lu Zhang performed the experiments, prepared figures and/or tables, and approved the final draft.

## Data Availability

The raw measurements are available in the Supplemental Files.

## Supplemental Information

Supplemental information for this article can be found online at http://dx.doi.org/10.7717/peerj.8726#supplemental-information.

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
