# Peer review of "Amendments to saline-sodic soils showed long-term effects on improving growth and yield of rice (Oryza sativa L.)"

_PeerJ, doi:10.7717/peerj.8726_

## Round 0.1 · original submission · Major Revisions

The article is of great interest because it addresses an important subject and, particularly, because it reports data from a long-term experiment (2009-2017).
However, the manuscript needs to be considerably improved before being published.
As both reviewers say, the English language needs substantial improvement. Reviewer 1 has done a nice work editing the manuscript. Even so, I recommend the final manuscript to be revised by an English-speaking person.
Both reviewers have made very helpful suggestions. Please take them into account.
Introduction
Please take into account the suggestions of reviewer 2 to improve the Introduction. I miss a rationale for the choice of the amendments tested.
Materials and Methods
The given information on the soil used in the study is scarce. The paragraph under "Soil physicochemical properties" should be moved to “Materials and Methods”. Moreover, it should be advisable to have some additional information: composition of the exchange complex (besides sodium), soil structure, soil classification (Soil Taxonomy or WRB) and, if possible, soil mineralogical composition. According to this, it must be specified why the soil is defined as a saline-sodic soil.
Please take into account the comments by both reviewers and my own in the annotated manuscript.
Results
Although data are from a long-term experiment, only data from year 2017 are presented and discussed. Taking for grant that sampling was carried out only in 2017, it would be great if you could have yield data from preceding years and see if there is a consistent trend with treatments.
From the Materials and Methods section, it is clear that results are from a field experiment. But several times in the manuscript the authors mention “panicles per pot”. Should it be “panicles per plot”?
See also the comments from both reviewers and those on my annotated manuscript.
Discussion
The discussion must be improved.
Results from the literature and from the present study are confused in the discussion.
The authors claim that results from the literature are similar to their own. But this often is not clear (lines 192, 196-199).
Some assertions do not coincide with the results (for example, line 237).
Some sentences are hardly intelligible (lines 192-194, 212-213, 221-222, 231-232, 233-235, 240-241).
Some sentences seem to be contradictory with each other (191 and 201-203). Please see the comments on the annotated manuscript.
Various different amendments were tested. The choice of these amendments must be justified in the Introduction. The effects of these amendments must be addressed separately in the Discussion.
Treatment M combines the three tested single treatments. The authors state that “M treatment is the best among the tested amendments”. Does this best performance of M treatment simply result from the addition of the beneficial effects of all three amendments or is there a synergy among amendments when they are applied together? Do you have any indication of this from your own results or from literature?

Reviewer 1 ·

Basic reporting

.

Experimental design

.

Validity of the findings

.

Additional comments

REVIEW (PeerJ Manuscript, 38932)

GENERAL COMMENTS:

This manuscript reports results from a study conducted in China to determine which
amendment treatment, among four tested, gave the best results in improving rice growth and
yield in a saline-sodic soil. The study is relevant because of the increasing need to expand food
production onto soils, like saline-sodic soils, that traditionally have been very unproductive. The
four treatments were compared against an untreated control and with each other by collecting
data on the osmotic potential in soil solution, the selective absorption (SA), selective transport
(ST), the distribution of K+ and Na+ and yield components in rice plants. The best results were
obtained when a combination of manure, sand, and gypsum was used as a soil amendment.

The study seems to have been conducted appropriately and the results will be of interest to
readers wanting information on the best way to treat saline-sodic soils to improve crop yields.
The writing, however, needs to be carefully reviewed and the entire manuscript needs editing.
Obviously, some of the problems are due to English probably not being the first language of the
authors.

I have written extensive edits directly on the hard copy pages of the manuscript. I have scanned
these edited pages into a .pdf file and returned this file with my review. I hope the authors find
these edits useful when revising their manuscript. Below, I make some specific comments about
things in the manuscript that I think need to also be addressed to improve the overall quality of
the manuscript.

SPECIFIC COMMENTS:

1. Title. The title can be shortened. The suggested shortened title still is informative and useful
to describe the paper.

2. Line 77-85. It is not clear whether the treatments were only applied once in 2017 and then
the soil and crop responses measured annually afterwards. Alternatively, were the same plots
treated every year with the same treatment. For example, was the gypsum plot treated every
year so that there were nine total applications of gypsum to the plots for this treatment before
sampling was done in 2017?

3. Since this study seems to have data on rice yields for the entire time of 2009 to 2017, I think
it is important to inform the reader as to the yield trends observed. For example, was the best
treatment in 2009 different compared to the best treatment in 2017? Were soil properties
quickly improved or did they improve slowly with treatment.

4. Line 98. When were the soils sampled? Were they sampled only in 2017 at planting? After
harvest?

5. Line 128 uses the term “concentration” and that is the correct term. But then many times
afterwards the term “content” is used. That is incorrect.

6. Lines 133-134. This sentence is not supported by the data. The differences in mean root Na+
concentrations were actually greater than the concentrations in shoots.

7. Lines 140-155. The writing in this section was not very clear. Please review and make
appropriate edits.

8. Line 174. Please report all data to only three significant digits. I very much doubt your
experiment was so precise as to report yields using six significant digits. For example, it is not
appropriate to report rice yield as being 8924.87 kg/hm2. Also, I believe the appropriate
abbreviation for hectare is “ha” and not “hm2”.

9. Lines 192-194. This sentence is not clear to me.

10. Lines 207-209. What is the meaning of “redundant Na+”? This sentence is not clear to me.

11. Line 212. What is the meaning of “bypass flow of rice”?

12. Line 232. Is it really true that increased Na+ uptake increased plant growth?

13. Font size should be increased in Figures 2 and 4. With computers, this is easy to do and
makes the figures easier to read.

14. Tables 1 and 2 need footnotes defining the treatments in the same way the treatments are
defined in the figures.

15. Table 2. Please restrict reporting of data to three significant digits. I don’t think it is possible,
in an experiment like this, to measure a difference of 0.01 kg/hm2 in rice yield. There are too
many sources of variation that are involved in this experiment to be so precise.

Annotated reviews are not available for download in order to protect the identity of reviewers who chose to remain anonymous.

Reviewer 2 ·

Basic reporting

In the present study, the long-term effects of different amendments application in saline-sodic soils on improving growth and yield of rice were analyzed. It is interesting and can provide a theoretical basis for the reuse of solid waste and reclamation of saline-sodic soils. I think that this paper is within the scope of the journal, but for the reason I detailed in the general comments I recommend to make a major revision before acceptance.

Experimental design

No comment.

Validity of the findings

No comment.

Additional comments

1. Your introduction needs more detail. The relevant review on reclamation of saline-sodic soils using desulfurization gypsum is not comprehensive, and the following references can help to improve this issue.
Potential flue gas desulfurization gypsum utilization in agriculture: A comprehensive review. Renewable and Sustainable Energy Reviews, 2018, 82:1969-1978.
Sodic soil properties and sunflower growth as affected by byproducts of flue gas desulfurization. Plos One,2012, 7(12): e52437.
Effect of byproducts of flue gas desulfurization on the soluble salts composition and chemical properties of sodic soils. Plos One, 2013, 8(8): e71011.
2.The English language should be improved to ensure that an international audience can clearly understand your text.
Some examples where the language could be improved include lines 40-41, 58 – the current phrasing makes comprehension difficult. The authors should check this issue throughout this manuscript.
3.The authors should describe field design and treatment in detail. The amendments were applied once or repeatedly. The layout of plots should be provied. The managment of plots should be added, including irrigation, etc.
4.Soil physicochemical properties in Section Results should be moved to Materials and Methods.
5.L121-122: The osmotic potential can serve as a good index for evaluating plant response to saline-alkali stress (Souza et al., 2012). should be moved to Materials and Methods.
6.L140-142: These contents should be moved to Materials and Methods.
7.L254: in the western Songnen Plain should be in the western Songnen Plain of China.

---

## Round 0.2 · Minor Revisions

I agree with both reviewers that the authors have made a good job revising the manuscript. I am attaching an annotated manuscript with minimal comments. I assume the suggestion of one of the reviewers that including a list of abbreviations would be helpful so I am returning the submission to you to give you a chance to address this if you wish.

Reviewer 1 ·

Basic reporting

No comments.

Experimental design

No comments.

Validity of the findings

No comments.

Additional comments

The authors have done a good job revising their manuscript taking into account the review comments. I have no problem accepting the manuscript as it is currently written. There are still some editorial issues, such as verb tense errors, etc. that will need to be corrected as the manuscript is set for publication.

The addition of Figure 6 helps make the paper stronger.

Reviewer 2 ·

Basic reporting

no comment

Experimental design

no comment

Validity of the findings

no comment

Additional comments

The mauscript has been improved, and it can be accepted for publication in present format.

---

## Round 0.3 · Minor Revisions

You have done a good job revising the manuscript. However, a few more details about the statistical analysis and experiment are needed. For the plant Na+ and K+ measurements, in which year(s) were these measured? For the ANOVA models, was experimental design (block, plot, etc) taken into account in the model? As fixed or random effects?

---

## Round 0.4 · Minor Revisions

We would still like you to provide more details for the ANOVA analysis in the methods section of the manuscript. As we understand the design, there are 3 replicate blocks, each split into 5 plots (or subplots) for the 5 treatments. In such a design one should have both treatment and block (and sometimes their interaction) as factors in the ANOVA model. It is unclear from your rebuttal letter whether this was done, and the manuscript methods states that a one-way ANOVA was used suggesting that block was not included in the model. Please provide, in the methods section, a description of the ANOVA model used in formula notation (e.g. yield ~ treatment + block). Please also provide as tables or supplementary tables, ANOVA tables that give the sum of squares, mean squares, degrees of freedom, F statistic, and p-value for the ANOVA analyses.

---

## Round 0.5 · accepted · Accept

I think that the effort of the authors, the required changes and the time spent have resulted in a substantial improvement of the article. So I am very glad to inform you that the article has finally been accepted.